# Arterial Blood Pressure, Neuronal Excitability, Mineral Metabolism and Cell Volume Regulation Mechanisms Revealed by *Xenopus laevis* oocytes

**DOI:** 10.3390/membranes12100911

**Published:** 2022-09-21

**Authors:** Gerardo Gamba

**Affiliations:** Molecular Physiology Unit, Instituto de Investigaciones Biomédicas, Universidad Nacional Autónoma de México and Instituto Nacional de Ciencias Médicas y Nutrición Salvador Zubirán, Mexico City 04510, Mexico; gamba@biomedicas.unam.mx

**Keywords:** *Xenopus laevis*, membrane proteins, salt transport, calcium sensing, hypertension, epilepsy

## Abstract

*Xenopus laevis* oocytes have been an invaluable tool to discover and explore the molecular mechanisms and characteristics of many proteins, in particular integral membrane proteins. The oocytes were fundamental in many projects designed to identify the cDNA encoding a diversity of membrane proteins including receptors, transporters, channels and pores. In addition to being a powerful tool for cloning, oocytes were later used to experiment with the functional characterization of many of the identified proteins. In this review I present an overview of my personal 30-year experience using *Xenopus laevis* oocytes and the impact this had on a variety of fields such as arterial blood pressure, neuronal excitability, mineral metabolism and cell volume regulation.

## 1. Introduction

The electroneutral cation-coupled chloride cotransporters (CCCs) is a superfamily of solute carriers (SLC12) membrane transporters involved in several aspects of human physiology that goes from cell volume regulation to modulation of neuronal excitability and arterial blood pressure [1]. The discovery, molecular identification and characterization of this family has been tremendously beneficiated by using *Xenopus laevis* oocytes as a tool to study and understand most of the members of the family. In this review I will tell the story of how the oocytes were essential in cloning the members of this family, and later, in defining many aspects of the transporter function and regulation.

The cation-coupled chloride cotransporters are solute carriers that translocate cations from one side of the membrane to the other, always coupled with chloride ions, following an stoichiometry of 1:1, not producing any change in membrane potential [1], hence the name of electroneutral.

The family is divided in two major branches. One composed by three members that use Na^+^ coupled to Cl^−^, with or without K^+^. Two of them are known as NKCC1 and NKCC2 (encoded by genes *SLC12A2* and *SLC12A1*, respectively) and perform as Na-K-2Cl cotransporters. NKCC1 is present in basically all cells of the body and in epithelial cells its expression is polarized to the basolateral membrane. In this case the function of NKCC1 is to provide the cells with the K^+^ or Cl^−^ ions to be secreted through the apical membrane, for example, the stria vascularis cells in the lateral wall of the cochlear duct of the inner ear that secrete K^+^ to the endolymph, or the airway epithelium in the respiratory tract that secretes Cl^−^ into the lumen. NKCC2 is exclusively expressed in the apical membrane of the thick ascending limb of Henle’s loop of the nephron, which is the functional unit of the kidney. Its function is crucial for salt reabsorption, blood pressure regulation, and calcium, magnesium, potassium, and water homeostasis. The activity of NKCC2 is responsible for the increased osmolarity of the interstitium of the renal medulla that is required for water reabsorption by the collecting duct when water conservation is required. NKCC2 is the target of loop diuretics, such as furosemide or bumetanide, which are worldwide used to induce natriuresis for the treatment of edematous states such as that observed in patients with heart, liver or kidney failure, and nephrotic syndrome, amongst other syndromes [1,2,3].

The third member of this branch is the Na-Cl cotransporter known as NCC (encoded by *SLC12A3*). NCC translocates Na^+^ and Cl^−^ following a stoichiomestry of 1:1. It is expressed in the apical membrane of the distal convoluted tubule, another part of the nephron that is critical for salt reabsorption and, thus, volume and blood pressure regulation. NCC is also part of the distal convoluted tubule K^+^-sensing mechanism, which plays a crucial role in K^+^ homeostasis [4,5]. This is because in the collecting duct, the most distal segment of the nephron, the K^+^ secretion through the K^+^ channel ROMK depends on the negative lumen generated by the electrogenic Na^+^ reabsorption by the apical Na^+^ channel ENaC. Thus, the most Na^+^ that is delivered to the collecting duct, the more the K^+^ that is secreted. Because the distal convoluted tubule is upstream the collecting duct, the activity of NCC determines the amount of Na^+^ delivered. A higher NCC activity (more Na^+^ reabsorption) results in less delivery of Na^+^ to the collecting duct, which in turn results in less K^+^ secretion. NCC is the target of the thiazide type diuretics that have been used for more than 60 years as a primary pharmacological treatment of arterial hypertension. There are some reports for the presence of NKCC2 or NCC outside the kidney, such as intestine [6], bone [7], pancreas [8] or dorsal ganglia [9], but their physiological role is still elusive. Interestingly, in the eel there is a second gene encoding for a Na-Cl cotransporter known as NCCβ that is nevertheless resistant to thiazide diuretics [10].

The other branch is composed by four members that moves K^+^ coupled to Cl^−^ and are known as K-Cl cotransporters, KCC, for which there are four different isoforms: KCC1 to KCC4, encoded by the genes SLC12A4 to SLC12A7). KCC1 is present in all cells of the body. KCC2 is exclusively expressed in neurons. KCC3 and KCC4 are present in many different epithelial and non-epithelial cells [11].

The major role of the cation-coupled chloride cotransporters is to move chloride ions to inside or outside the cell. Because these cotransporters use the driving force generated by the Na-K-ATPase, the influx of Na^+^ or efflux of K^+^ by the sodium- or potassium-driven branches, respectively, is resolved by the Na-K-ATPase and thus, the net result of their activity is the increase, or decrease, in the intracellular chloride concentration ([Cl^−^]_i_). NKCC1/2 and NCC increases [Cl^−^]_i_, while the KCCs decrease [Cl^−^]_i_. The net movement of chloride by the CCCs has implications for the cell volume regulation, neuronal excitability and transepithelial salt transport.

## 2. *Xenopus laevis* Oocytes Were Crucial for the Molecular Identification and Characterization of the CCCs

At the end of the 1980s, the existence of the CCCs was already known by many physiological studies; however, the sequence of the genes was unknown [12]. There was also no possibility to raise antibodies against them. Thus, the advance of the field was very restricted. Following the expression cloning strategy using the heterologous expression system of *Xenopus laevis* oocytes (Figure 1) we were able to identify at the molecular level the cDNA encoding NCC from the winter flounder’s urinary bladder [13]. Because the cotransporters are electroneutral, the two-voltage clamp technique was not useful for this project. We began by assessing the chloride-dependent and thiazide-sensitive ^22^Na^+^ uptake in oocytes microinjected with mRNA extracted from the winter flounder’s urinary bladder (Figure 1). We chose this tissue because it was previously shown that the Na-Cl cotransporter was present here [14], was sensitive to thiazides [15] and constituted a richer source of mRNA, as opposed to the kidney, where the Na-Cl cotransporter is only present in the distal convoluted tubule that constitutes probably less than 5% of all renal cells. Once we corroborated the high expression of the putative cotransporter in the mRNA injected oocytes, we constructed a cDNA library and analyzed pools of clones from which we extracted the cDNA and synthetized cRNA in vitro (Figure 1). Once that the cRNA obtained from a pool of clones induced the expression of the cotransporter, then it was just a matter of time to analyze individual clones, until we finally obtained a single clone whose cRNA induced in oocytes the presence of a chloride-dependent and thiazide-sensitive Na-Cl cotransporter (Figure 1). The sequence revealed an open reading frame of 3069 base pairs encoding a 1023 amino acid membrane protein. This is how, using *Xenopus laevis* oocytes, NCC was the first member of the SLC12 family to be identified at the molecular level [13].

The molecular identification of the flounder’s NCC cDNA allowed us to use that probe to identify, using southern blot techniques, the cDNA encoding not only the rat and mouse NCC, but also the rat and mouse NKCC2 cDNA [16]. The sequences of NCC and NKCC2 were used to design degenerative PCR primers which were instrumental in identifying also the mouse cDNA encoding the NKCC1 cDNA [17]. The functional characterization of these three cotransporters was achieved in *Xenopus* oocytes and since then many aspects of the function and regulation have been obtained using this expression system, particularly for NCC and NKCC2, which for unclear reasons, it has been impossible to obtain a robust expression using mammalian cells [18].

Mutations in *SLC12A2*, encoding NKCC1, have been associated with a rare genetic syndrome known as Kilquist syndrome featuring sensorineural deafness, intestinal and respiratory dysfunction, neuropsychological delay, and severe xerostomia (autosomal recessive; loss of function) [19]. The complexity of this syndrome in consistent with the many features observed in the NKCC1 knockout mice [20] (Figure 2).

The cDNA encoding for the KCCs were identified during the 1990s by in silico approaches, taking the advantage of the expressed sequence tags, which at that time were deposited daily in genome data bases because of the human genome project. KCC1 was first identified by Forbush and coworkers and was functionally characterized using HEK-293 cells [21]. KCC2 was identified by homology with KCC1 and recognized as a neuronal specific isoform [22] and the functional characterization was obtained later in *Xenopus* oocytes [23]. The KCC3 and KCC4 isoforms were also identified from mouse and human cDNA libraries using expressed sequence tags that exhibited some degree of homology with KCC1 and KCC2 and the functional characterization was obtained using *Xenopus* oocytes [24]. KCC3 was cloned simultaneously by another group using a different strategy [25]. Thus, the era of the molecular studies of the SLC12 family of electroneutral cation-chloride cotransporters began in great part thanks to the use of *Xenopus* oocytes as a heterologous expression system.

## 3. The CCCs Are Involved in Neuronal Excitability, Cell Volume and Blood Pressure Regulation

As mentioned before, the CCCs activity is critical to define the [Cl^−^]_i_. Work performed in *Xenopus* oocytes and in mammalian expression cells demonstrated that the activity of the two branches of the CCCs is inversely regulated by phosphorylation/dephosphorylation processes. The Na^+^-driven cotransporters NKCC1, NKCC2, and NCC activity is promoted by phosphorylation of key residues in the amino terminal domain and decreased by dephosphorylation [26,27,28,29]. In contrast, KCCs activity is reduced by phosphorylation of key residues in their carboxyl-terminal domain and activated by protein phosphatases [30,31]. Thus, in each cell expressing for instance NKCC1 and any of the KCCs, phosphorylation of the cotransporters results in activation of NKCC1 and inhibition of the KCCs, increasing the [Cl^−^]_I_; whereas dephosphorylation inhibits NKCC1 and activates the KCCs, resulting in a decrease in [Cl^−^]_i_.

The kinase directly responsible for the CCCs phosphorylation are the STE-20 proline alanine rich kinase and/or the oxidative stress responsive kinase 1 known as SPAK or OSR1, respectively [32,33]. These kinases are in turn under control of the with-no-lysine kinases (WNKs) that directly phosphorylate SPAK/OSR1 to increase their activity towards CCCs [34]. There are four genes encoding isoforms of the WNK kinases, WNK1 to WNK4 [35]. Intronic deletions of intron 1 of WNK1 and missense mutations in WNK4 in a particularly high conserved acidic region between WNKs results in Familiar Hyperkalemic Hypertension syndrome [36]. *Xenopus* oocytes helped to reveal that WNKs modulate the activity of the CCCs by promoting the activating phosphorylation of NKCC1/2 and NCC and the inhibiting phosphorylation of KCCs [30,37,38,39,40,41,42,43,44]. The activation of WNKs occurs by autophosphorylation and to date, it is clear that at least WNK1 and WNK4 are sensitive to [Cl^−^]_i_ [38,45]. If [Cl^−^]_i_ is increased, the activity of WNK1 and WNK4 is reduced since Cl^−^ binds directly to the WNKs, preventing their activation. WNK4 is the most sensitive kinase to Cl^−^ [46] and that is why WNK4 is the key regulatory kinase of NCC in the distal convoluted tubule, where the presence of any other WNK besides WNK4 would result in constitutive activation of NCC, as occurs in the FHH syndrome due to intronic deletion of WNK1, in which the consequence of the intronic deletion is that WNK1 is ectopically expressed in the distal tubule, remaining active at the normal [Cl^−^]_i_ of the tubular cells [39]. Therefore, the sensitivity of WNK1 and WNK4 for [Cl^−^]_i_ represents a classical feedback mechanism. If [Cl^−^]_i_ is decreased, WNKs are activated, promoting phosphorylation of SPAK/OSR1, activating the Na^+^-driven CCCs, but inhibiting de K^+^-driven members of the family and increasing [Cl^−^]_i_ that in turns prevents the activation of WNKs.

### 3.1. Neuronal Excitability

The modulation of the [Cl^−^]_i_ by the CCCs, particularly NKCC1 and KCC2 has been shown to be responsible for the differential effect of the neurotransmitter GABA for whom the target is a membrane receptor coupled to a chloride channel. If neurons express an excess of NKCC1 activity over KCC2, as for instance in the embryonic neurons, the [Cl^−^]_i_ will be above the potential equilibrium resulting in the opening of the chloride channel, chloride efflux and depolarization. In these circumstances GABA performs as an excitatory neurotransmitter. In contrast, in neurons with higher activity of KCC2 over NKCC1, as occurs after birth, the [Cl^−^]_i_ will be below the potential equilibrium and the GABA effect of opening chloride channels results in chloride influx, hyperpolarization, and inhibition of neuron activity. In these circumstances GABA performs as an inhibitory neurotransmitter [47,48,49]. For instance, the KCC2 knockout mice in which only about 5% of KCC2 is active develops an almost permanent epilepsy and is lethal in the first 17 days [50]. Loss-of-function mutations in one allele of *SLC12A5,* encoding KCC2, produce a severe infantile-onset pharmaco-resistant epilepsy syndrome, epilepsy of infancy with migrating focal seizures (EIMFS) [51,52]. This is a key model of regulation of different pathways in the central nervous system, such as the circadian rhythm in which one pathway can be excitatory at certain hours and inhibitory in others [53]. The CCCs regulation of [Cl^−^]_i_ and its effect on GABAergic potentials is one of the major advances in the understanding of neuronal function regulation and it was due in part by experiments performed in *Xenopus* oocytes expression system. As a final example, *Xenopus* oocyte expression system allowed the identification and characterization of mutation in *SLC12A6,* encoding for KCC3. Mutations in this membrane transporter cause a very complex neurological disease known as Peripheral neuropathy associated with agenesis of the corpus callosum with mental retardation and sensorimotor neuropathy [54].

### 3.2. Cell Volume Regulation

The activity of CCCs in all cells plays a crucial role in cell volume regulation when extracellular osmolarity is changed. An increase in osmolarity outside the cells promotes water efflux, resulting in cell shrinkage that prompts a response known as regulatory volume increase (RVI). In this response, cells activate the influx of ions to equalize the intracellular osmolarity to that of outside the cell and thus restore normal volume. RVI is accomplished in part by activating the NKCC1 cotransporter. In contrast, a decrease in extracellular osmolarity produces water influx and thus cell swelling, activating a response known as regulatory volume decrease (RVD), in which ions must leave the cell to reach an osmolarity similar to that of outside. Here, the K-Cl cotransporters, particularly KCC1, play and important role. During RVI the CCCs are phosphorylated activating NKCC1 and inhibiting KCC1 while, during RVD they are dephosphorylated promoting the NKCC1 inhibition and KCC1 activation [55,56].

The regulation of CCCs during cell volume changes seems counterintuitive with the modulation of the cotransporters by changes in the [Cl^−^]_i_ because during cell shrinkage the [Cl^−^]_i_ is expected to be increased; however, the NKCC1 is activated, while during cell swelling, the [Cl^−^]_i_ is expected to be decreased, but the KCCs are activated. The possible explanation for this discrepancy could be that different WNK kinases modulate the CCCs during cell volume changes. We observed in oocytes that WNK3 is a powerful activator of NKCC1 [44] and inhibitor of KCCs [30], bypassing the tonicity requirements for activation/inhibition of this cotransporters. For instance, co-expression of NKCC1 with wild-type WNK3 results in activation of the cotransporter, even exposing the cells to hypotonicity in which in the absence of WNK3 is inhibited [44]. In contrast, co-expression of KCCs with the catalytically inactive WNK3 results in activation of KCCs, even without the need of cell swelling, suggesting that by preventing WNK3 activity with a dominant negative inactive form, KCCs are activated despite no changes in cell volume [30]. We further observed in HEK-293 cells that WNK3 and its catalytically inactive form indeed modulate the RVD [57], and more recently, using oocytes, that WNK3 is not sensitive to changes in the [Cl^−^]_i_. We found that WNK3 can activate the family of CCCs even at very high [Cl^−^]_i_ (about 50 mM) [43,58], in contrast to WNK1 and WNK4, who are Cl^−^-sensitive kinases [37]. We observed in *Xenopus* oocytes that changes in cell volume are the modulatory signals that switch on and off phosphorylation of WNK3, but not that of WNK4. Cell swelling reduces WNK3 phosphorylation while cell shrinkage increases it [37].

### 3.3. Arterial Blood Pressure

A third role for CCCs, particularly, NCC and NKCC2 that was revealed at least in part by experiments performed in *Xenopus* oocytes is the regulation of arterial blood pressure. Inactivating mutations of NKCC2 and NCC are the cause of Bartter syndrome type I and Gitleman’s syndrome, characterized by hypokalemic metabolic alkalosis with a reduction in blood pressure due to renal salt wasting. The first studies demonstrating the effect of mutations in the cotransporters function and activity were performed in *Xenopus* oocytes [59,60]. Conversely, as mentioned before, mutations in WNK1 and WNK4, as well as in two ubiquitin ligase complex proteins known as CUL3 and KLHL3 produce FHH featuring hyperkalemic metabolic acidosis with arterial hypertension. Work in oocytes helped to understand that WNKs, particularly WNK4, is a chloride-sensitive kinase whose activity is reduced when the [Cl^−^]_I_ rises and is increased with the [Cl^−^]_i_ decrease [38,45]. When active, WNK4 phosphorylates SPAK/OSR1, that in turn phosphorylates NCC, increasing its activity [34,61]. The overactivation of NCC results in more salt reabsorption, hypertension and reduced renal capacity for K^+^ excretion, consequently causing hyperkalemia [62].

No disease has been associated yet with mutations in the SLC12A7 gene encoding KCC4. However, elimination of this gene in mice results in two interesting consequences [63]. One the one hand, sensorineural deafness because KCC4 in the basolateral membrane of the stria vascularis cells is critical to reduce the [Cl^−^]_i_ to keep constant the activity of NKCC1 that provides the K^+^ to be secreted to the endolymph. On the other hand, the mouse model has shown the development of renal tubular acidosis because of a similar mechanism. KCC4 is required in basolateral membrane of intercallated cells in the collecting duct, which are the cells that secrete H^+^ into urine, to keep the [Cl^−^]_i_ low to allow the interchange of intracellular HCO_3_^−^ with extracellular Cl^−^ and the H^+^ is secreted into urine. Interestingly, a recent work from our group, in which oocytes were used among many other techniques, demonstrated that sirtuin 7 increases the half-life of KCC4 by preventing its acetylation and that elimination of sirtuin 7 expression in the mouse kidney is associated with reduction in KCC4 expression and mild tubular acidosis [64].

## 4. The Birth of the Calcium-Sensing Field in Mineral Metabolism Was Possible Thanks to *Xenopus laevis* Oocytes

Another exciting subject of medicine and biology in which I had the opportunity to participate was a seminal work of the discovery and identification of the cDNA encoding the calcium-sensing receptor (CaSR) [65]. It was known for years that extracellular calcium concentration within a very narrow physiological range modulates the secretion of the parathyroid hormone (PTH). At 1.0 mM Ca^2+^ concentration PTH secretion is maximal while at 1.3 mM it is completely prevented [66,67]. The physiological role of PTH is to increase calcium concentration by releasing calcium from the bone, increasing renal reabsorption of Ca^2+^ and promoting activation of vitamin D_3_, which in turn increases the gastrointestinal absorption of Ca^2+^. Thus, a classical feedback mechanism occurs. PTH increases serum Ca^2+^ concentration, which in turns prevents PTH secretion.

The molecular mechanism by which calcium ions inhibits PTH secretion was not known. It was observed at the end of the 1980s that the Ca^2+^ induced inhibition of PTH secretion was accompanied by phosphoinositide turnover and cytosolic Ca^2+^ release and that this effect could be reached also with other divalent or trivalent cations, as well as, by polyvalent cations such a neomycin, suggesting a mechanism involving a membrane receptor, rather than a channel [68]. By that time, we were already working on the cloning of the NaCl cotransporter from winter’s flounder bladder when Edward Brown approached us with the proposal to explore the system to identify the putative receptor. We used bovine parathyroid glands as a source of mRNA. Because the proposed mechanism was that the activation of the putative receptor would produce phosphoinositides that increase intracellular Ca^2+^, depolarizing the cell, the consequence will be the opening of an endogenous chloride channel of the oocyte to balance the charges. Thus, it was the chloride channel activity what we used to follow the expression of the receptor [65]. It took us about a year to finally obtain and sequence a single 5.3 kb cDNA clone containing a 3255 pb open reading frame encoding a 1085 amino acid membrane protein that contains a large extracellular domain, seven transmembrane spanning regions and an intracellular carboxy terminal domain. A classical structure of the G-coupled receptors. The new receptor was named as the Calcium-Sensing Receptor (CaSR) and it was shown to be present in many tissues.

The cloning of the CaSR opened a new field in medicine and biology [69,70]. It was later demonstrated that the CaSR is present in basically all cells and from bacteria to humans. Mutations in this gene are the cause of Familial Hypocalciuric Hypercalcemia, a dominant, inherited, very mild, almost only biochemical disease, and recessively inherited severe neonatal hypoparathyroidism that requires the surgical extraction of the parathyroid glands in the first week of life to prevent infant death. Both diseases are due to inactivating mutations of the CaSR. In the first case, only one allele is mutated, and in the second one, both alleles are inactive. Activating mutations of the CaSR result in a disease known as autosomal dominant hypocalcemia.

The cloning of the CaSR allowed not only to understand many pathophysiological mechanisms in a diversity of organs, but also the development of calcimimetic drugs (allosteric modulators type II, such as cinacalcet) which increase the sensitivity of the receptor for calcium, reducing PTH secretion. Cinacalcet is now a standard treatment for the secondary hyperparathyroidism that occurs in patients with end stage renal disease [71]. The field evolved so quickly that ten years after the initial cloning of the CaSR, an editorial in the New England Journal of Medicine mentioned that the history of the calcium-sensing receptor serves as the paradigm of translational research [72]. A recent review accounts for the incredible growth of this field in the last 30 years [70]. All this was possible in the first place thanks to the *Xenopus laevis* oocytes.

## 5. Conclusions

*Xenopus laevis* oocytes have been a powerful tool for the discovery and identification of membrane proteins. The field of electroneutral cation chloride cotransporters advanced tremendously due to the identification of cDNA encoding members of the family using oocytes. These cotransporters have important implications in the regulation of arterial blood pressure, neuronal excitability, and cell volume. The discovery of the calcium-sensing receptor was possible thanks to the oocytes and opened an important field in the mineral metabolism physiology.

## Figures and Tables

**Figure 1 membranes-12-00911-f001:**
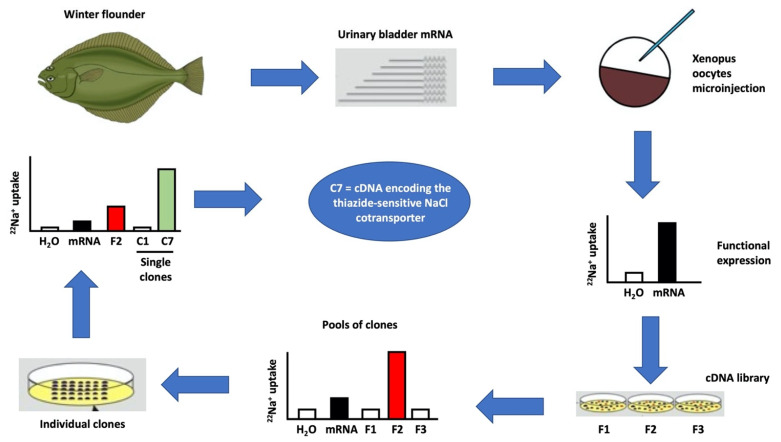
Expression cloning of the NaCl cotransporter, NCC. mRNA extracted from the winter flounder urinary bladder was injected into Xenopus oocytes and three days later the thiazide-sensitive and Cl^−^-dependent ^22^Na^+^-uptake was assessed. Once the expression was established, a cDNA from the urinary bladder was constructed and filters containing about 500 clones each were prepared. Pooled bacteria for each filter were used to extract their cDNA to in vitro transcribe cRNA that was used to inject oocytes and search for NCC activity. As expected, the level of expression from the pooled bacteria was higher than from the total mRNA. Then, cloned from the positive filter were studied by groups until the cRNA transcribed from a single individual clone (C7) induced the presence of the thiazide-sensitive and Cl^−^-dependent ^22^Na^+^-uptake, which as expected was higher than that observed from the pooled clones. C7 clone cDNA encoded for NCC [13].

**Figure 2 membranes-12-00911-f002:**
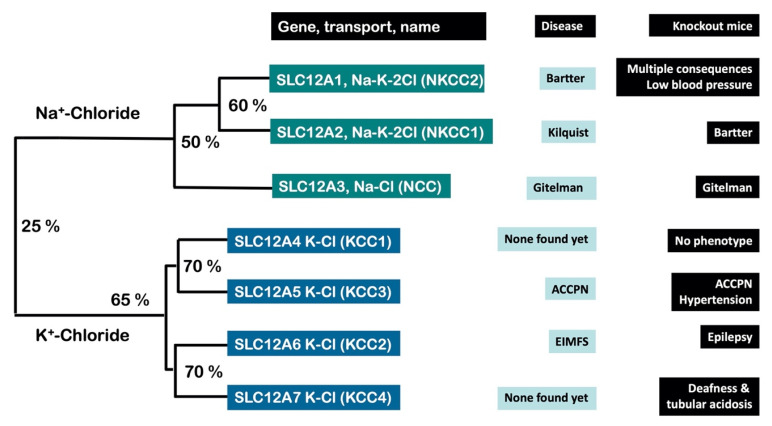
The SLC12A family of electroneutral cation chloride cotransporters. Seven members composing the Na^+^-driven and the K^+^-driven branches are shown, but there are two most distant members SLC12A8 and SLC12A9 not shown. The figure depicts the gene name, the cotransporter’s behavior, the human disease produced by inactivating mutations and the phenotype of the knockout mice. The percentages depict the molecular identity among members of the family. ACCPN = agenesis of the corpus callosum polyneuropathy. EIMFS = epilepsy of infancy with migrating focal seizures.

## Data Availability

Not applicable.

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
