# Peer review of "Arterial Blood Pressure, Neuronal Excitability, Mineral Metabolism and Cell Volume Regulation Mechanisms Revealed by Xenopus laevis oocytes"

_membranes, 2022, doi:10.3390/membranes12100911_

Round 1

Reviewer 1 Report

Xenopus oocytes is a conventional model system that has long been used in characterising  ion channels and  transporters. The review in its current form is more of a compilation of the authors own work with limited literature survey and critical observation/assessment of the previous work done using the model of interest.  There are a number of major limitations:

1) The manuscript does not have precise scope and the title is too broad it should reflect the type of transporters/receptors covered in this review which is CCCs and CaSR.

2) The manuscript is mostly a summary of the author’s own work with little mention of  relevant studies from other groups where Xenopus oocytes were used as a model system for characterizing CCCs and CaSR.

3) The advantages and limitations of Xenopus oocytes are not outlined and compared to other current systems that are in use for studying ion channels or in this case CCCs and CaSR

4) The manuscript does not provide new insight into the topic through innovative thinking or stimulating debate. Nor does it address current knowledge gaps and recommends future research direction.

Specific comments

1) Citation of references is missing in a number of occasions for example (lines 34-63, 278-284 and 291-303)

2) Figure 2: it is not clear what the percentages in the diagram represent 

 3) lines 57-59 ” This is because Na+ delivery to the collecting duct, which is  the most distal segment of the nephron, defines the rate of K+ secretion, therefore the Na+that NCC manages to recover impacts the rate of K+ secretion” This sentence requires revision 

4) lines 102-104 “This is how, using Xenopus laevis oocytes, NCC was the first member of the SLC12 family to be identified at the molecular level [9].” This sentence requires revision

5) The manuscript would benefit from a round of proof reading

Author Response

The reviewer is right in his/her general comments 1 to 4. The manuscript was not written to discuss the advantages or limitations of the Xenopus laevis oocytes as a tool or to provide an innovative thinking or stimulate the debate and that the manuscript is mostly a summary of the author’s won work…... In the abstract it is stated: “In this review I present an overview of my personal 30 year experience using Xenopus laevis oocytes and the impact this had on a variety of fields such as arterial blood pressure, neuronal excitability, mineral metabolism and cell volume regulation”.  So, the idea of the review is to highlight, by using my own experience and history, how the Xenopus oocytes were instrumental in generate knowledge and tools that have been of most importance for physiological and medical areas. I believe the scope is to let the reader know to what extent the oocytes were a key tool in these areas.  

Specific comments

1) I did not include initially references for these lines because it is common knowledge, but following the reviewer’s suggestion I have added references about it.

2) Sorry for missing the explanation. It has been added to the figure legend.

3) The line has been rewritten.

4) The line has been rewritten.

Reviewer 2 Report

Nice paper. It should be published.

Author Response

I thank the reviewer for his/her positive comment.

Reviewer 3 Report

In the references, some have both volume and issue numbers. Some only have volume numbers. To keep it consistent, I would suggest to remove issue numbers of the journal, just keep the volume numbers. 

Author Response

Thank you for notice this mistake and sorry for that. The reference list was corrected.

Round 2

Reviewer 1 Report

The raised concerns were not addressed satisfactorily. Writing a summary of one's experience in a certain topic will be more complete and meaningful only if it was placed in context with the work of others in the field and with relevant system models, something that had not been incorporated in the revised version